

# Predictors of periodontal and caries related perinatal oral healthcare, investigation of dentists' practices: a cross-sectional study

Muhammad Qasim Javed[1], Usman Anwer Bhatti[2], Arham Riaz[3] and Farooq Ahmad Chaudhary[4]

[1] Department of Conservative Dental Sciences and Endodontics, College of Dentistry, Qassim University, Buraydah, Qassim, Saudi Arabia
[2] Department of Operative Dentistry, College of Dentistry, Riphah International University, Islamabad, Pakistan
[3] Community Dentistry, Academy of Continuing Health Education and Research, Islamabad, Pakistan
[4] School of Dentistry, Shaheed Zulfiqar Ali Bhutto Medical University, Islamabad, Pakistan

Corresponding author
Muhammad Qasim Javed,
M.Anayat@qu.edu.sa

## ABSTRACT

**Background:** The objectives of the study were to assess the knowledge, attitude, and practice of dentists towards providing oral health care to pregnant women and to identify barriers and predictors of periodontal and caries related perinatal oral healthcare practices.

**Methods:** A cross-sectional analytical survey was conducted on dentists by using a random sampling technique, and a pre-validated questionnaire was delivered to 350 dentists from May 2018 to October 2018. Data were analyzed by utilizing SPSS software. Frequencies and percentages were recorded for descriptive variables. Binary logistic regression was used to analyze the probability of predicting group membership to the dependent variable using different independent variables determined from contingency tables.

**Results:** Overall response rate was 41%. The mean knowledge score of respondents was 15.86 ± 3.34. The lowest correct responses were noted in the questions related to periodontal health. It was found that the advice to delay dental visits until after pregnancy was eight times more likely to be observed among dentists who lacked the knowledge of importance of oral health during pregnancy ($P = 0.04$, OR = 8.75). Dentists were more likely to consult obstetricians regarding dental procedures when they fear a risk of labor in the dental practice ($P < 0.05$, OR = 3.72). Dentists who had the knowledge of periodontal disease association with preterm delivery were about four times more likely to treat periodontal disease during pregnancy ($P = 0.01$, OR = 3.95). Dentists knowing the association between maternal oral health and childhood decay were more likely to counsel pregnant patients regarding caries prevention ($P > 0.05$, OR = 3.75).

**Conclusions:** Collectively the results indicated few gaps in knowledge among some dentists and a need to improve existing attitudes towards perinatal oral health. Dentists failing to recognize the importance of perinatal oral health are more likely to be hesitant in treating pregnant patients. Failing to recognize the link between periodontal disease and obstetric complications increases the possibility of hesitance to counsel pregnant patients regarding the same. The appreciation of the evidence for

poor perinatal oral health and risk of early childhood caries increases the likelihood of counseling by dentists on caries prevention.

# INTRODUCTION

Pregnancy is a physiologically challenging condition with numerous hormonal changes contributing to an increased risk of oral diseases (*Riaz et al., 2020*; *Krüger et al., 2015*). The two oral diseases responsible for most oral health problems in pregnant women are caries and periodontal diseases (*Riaz et al., 2020*; *Krüger et al., 2015*; *Mobeen et al., 2008*). In a developing country like Pakistan, moderate to severe oral disease burden has been reported concerning pregnant women (*Mobeen et al., 2008*). Preterm low birth weight (LBW), stillbirth, neonatal, and prenatal death are some of the serious implications of inadequate periodontal health of pregnant women (*Vergnes & Sixou, 2007*; *Wazir et al., 2019*; *Loesche, 1997*). LBW is a serious health care problem in Pakistan where the reported incidence of LBW is 37% (*Qureshi et al., 2005*). Moreover, LBW can contribute to early childhood caries (*da Silva Castro et al., 2019*). Accordingly, WHO noted preterm birth (PB) (delivery at < 37 weeks gestation) and low birth weight as the two most common reasons for neonatal mortality (*WHO, 2012*). Current evidence base suggests that PB usually leads to LBW (baby weight < 2.5 kg) (*Gupta et al., 2015*; *Scannapieco, Bush & Paju, 2003*). *Boggess & Edelstein (2006)* attributed almost 18% of the PB and LBW to periodontal disease. This might be the result of increased levels of inflammatory mediators and cytokines (IL-8, TNF) released during periodontal inflammation. Furthermore, the mediators and cytokines may enhance prostaglandin production resulting in early labor (*Bokhari & Khan, 2006*).

The oral health of the expectant mothers has been indicated as a significant predictor of children's oral health, and mothers are usually considered the primary source of cariogenic bacteria transmission to children (*Boggess et al., 2011*). Researchers have reported similarities (genotypic and phenotypic) among the streptococcus mutans (SM) bacteria found in the oral cavity of the child and mother (*Meyer, Geurtsen & Günay, 2010*). Considering this, children with high SM count during early life might have it transmitted from the saliva of a mother with an active carious lesion, particularly during certain feeding practices (*George et al., 2011*; *Duff et al., 2017*). Moreover, studies show that maternal oral hygiene practices and diet can also influence this risk (*Proença et al., 2015*). Hence, maintaining the oral health of pregnant women is not only important for minimizing the risk of obstetric complications but also for the future oral health of both mother and infant (*Finlayson, Gupta & Ramos-Gomez, 2017*).

The susceptibility of pregnant women to problems pertaining to oral health is further increased as only half of these women are reported to seek oral health care (*Geisinger et al., 2019*). Considering that most dental care is safe during pregnancy, such a low number of women seeking care is alarming (*Finlayson, Gupta & Ramos-Gomez, 2017*).

Lack of care can lead to an escalation of pre-existing dental problems with the literature reporting a high prevalence of dental pain among pregnant women (*Krüger et al., 2015*).

There can be different barriers to seeking oral health care among pregnant women, like economics and concern for fetal health. However, in addition to patient-related factors, there is a matter of dentists' perception which acts as a barrier in providing oral health care to pregnant women (*Vieira et al., 2015*; *Lee et al., 2010*). Literature highlights the need for improving the knowledge of dentists with regards to the provision of oral health care to pregnant women (*Vieira et al., 2015*; *Lee et al., 2010*; *George et al., 2017*; *Mayberry, Norrix & Farrell, 2017*; *Huebner et al., 2009*). There have been only a few studies investigating the perception of dentists in Pakistan towards the provision of oral health care to pregnant women (*Wali et al., 2016*). In light of the observed association of oral disease severity with neonatal mortality (*Mobeen et al., 2008*), it is important to explore the knowledge, attitudes, and perceptions of Pakistani dentists regarding the oral healthcare of pregnant women.

Therefore, the objectives of this study were to assess the knowledge, attitude, and practice of dentists towards providing oral health care to pregnant women, and to identify barriers and predictors of periodontal and caries related to perinatal oral healthcare practices.

## MATERIALS & METHODS

A cross-sectional analytical survey was conducted from May 2018 to October 2018 on the dental practitioners registered with Pakistan Medical Commission and are working in the Islamabad-Rawalpindi metropolitan area, Pakistan by utilizing a convenience sampling technique. The participant information sheets, questionnaires, and consent forms were delivered by mail to 350 dentists along with the stamped envelopes for returning the filled questionnaires and consent forms by post. Potential participants were informed that participation was voluntary, and written informed consent was taken from all the participants. The researchers kept all the obtained information confidential. Ethics approval was acquired from Riphah International University's Institutional Review Board (IIDC/IRC/2018/03/07).

The validated questionnaire developed by *George et al. (2017)* was used with permission. The survey questionnaire had four subsections with questions on knowledge, attitude, practices concerning oral healthcare during pregnancy, and barriers perceived for providing oral healthcare to pregnant women. Moreover, demographic details were also acquired. The questionnaire design comprised of multiple-choice items ("true"/ "False"/"Not sure") in the domain of knowledge and for the assessment of dentists' attitudes, practices, and barriers Likert scales were used.

The population prevalence of counseling pregnant patients regarding caries prevention and transmission was unknown. Consequently, 50% prevalence was presumed for the conservative estimate. For estimation of the population proportion with the power value of 80 percent, alpha ($\alpha$) of 0.05, and precision of +0.1, the sample size of 97 was calculated.

Survey data were analyzed by utilizing SPSS version 23 (SPSS, Chicago, IL, USA). Descriptive statistics were utilized for assessing the knowledge, practices, attitudes, and

barriers to practicing dentists concerning oral healthcare during pregnancy. The questions' responses in the domain of knowledge were calculated out of 25 after marking the responses to the questions against the systematic reviews (*Ide & Papapanou, 2013*; *Leong et al., 2013*) and evidence-based guidelines (*California Dental Association Foundation; American College of Obstetricians and Gynecologists, District IX, 2010*; *American Academy of Pediatric Dentistry, 2011*). To explore the knowledgeability, attitudes, practices, and barriers of dentists regarding oral health in pregnancy, descriptive statistical analyses were used.

Contingency tables were generated to explore the relationship between different dependent variables of interest. Those variables expressed on the ordinal scale were converted into dichotomous variables. For instance, the items on the practice domain which were originally expressed on a three point scale, comprising of "always", "sometimes" and "never" responses, were dichotomized by combining responses of "always" and "sometimes" and leaving "never" as a separate category. Similarly, items for the attitude and barrier domain, which were originally expressed on a five-point Likert scale were dichotomized by merging responses of "strongly agree" and "agree" as one category and combining the remaining responses as another. Chi-square and Fisher's exact test were used to analyze associations between different categorical (knowledge, attitude, practice, and barrier) variables of interest. Those variables which exhibited a statistically significant association were then selected for running a regression.

Binary logistic regression was used to analyze the probability of predicting group membership to the dependent variable using different independent variables determined from contingency tables. Two variables of the practice domain (counseling pregnant women regarding caries prevention and transmission, counseling regarding the association of poor perinatal periodontal health and negative pregnancy outcomes) and one from the attitude domain (treatment of periodontal disease during pregnancy positively affects pregnancy outcome) were identified as the dependent variables to generate prediction models, while various items from the knowledge domain were identified as independent (predictor) variables. These dependent and independent variables were treated in their dichotomous form and not in their ordinal form. The goodness of fit of the prediction models was determined using the Hosmer and Lemeshow Test, with a $P$-value of $> 0.05$ indicating a good fit of the model compared to the null model. Odds ratios with 95% confidence intervals were calculated for each of the independent (predictor) variables.

## RESULTS

Out of 350 dentists, 196 responded. After excluding 43 incomplete questionnaires, a total of 143 were selected for final analysis, with a response rate of roughly 41%. The mean age of the respondents was $30.64 \pm 5.1$ years with a mean experience of $6.2 \pm 4.3$ years and about $36.9 \pm 16.7$ working hours per week. The majority of the respondents were female (58%) working in a private setting (69.2%) with a bachelor's degree (58.7%). The average numbers of pregnant patients seen by these dentists per month were between one and five (72%).

**Table 1 Percentage of correct responses to perinatal oral health knowledge items (N = 143) of dentists.**

| Item (correct response as per current evidence and guidelines) | Correct response N (%) |
|---|---|
| Pregnancy exacerbates existing dental problems (true) | 119 (83.2) |
| Gingivitis is more serious than periodontitis (false) | 116 (81.1) |
| Calcium will be drawn out of mother's teeth by developing baby (false) | 116 (81.1) |
| Gingivitis is a potentially reversible infection of the gums (true) | 132(92.3) |
| Poor maternal oral health can contribute to early childhood decay (true) | 58 (40.6) |
| Periodontal disease has been associated with the following: | |
| Stillbirth (true) | 24 (16.8) |
| Preterm delivery (true) | 51 (35.7) |
| Spontaneous abortion/miscarriage (true) | 39 (27.3) |
| Preeclampsia (true) | 30 (21) |
| Low birth weight (true) | 62 (43.4) |
| Women should receive preventive dental care during pregnancy (true) | 113 (79) |
| Basic dental treatment is safe during pregnancy (true) | 127(88.8) |
| It is unsafe to obtain dental radiographs in pregnant women (false) | 45 (31.5) |
| Pregnant women should receive only emergency dental care (false) | 35 (24.5) |
| Elective dental treatment should be delayed until after pregnancy (true) | 106 (74.1) |
| These dental procedures are safe during pregnancy: | |
| Extractions (true) | 70 (49) |
| Local anaesthetic (true) | 74 (51.7) |
| Root canal treatment (true) | 97 (67.8) |
| Scaling and root planning (true) | 116(81.1) |
| These medications are safe during pregnancy: | |
| Paracetamol (true) | 132 (92.3) |
| Aspirin (false) | 128 (89.5) |
| NSAIDs (false) | 119 (83.2) |
| Amoxicillin (true) | 104 (72.7) |
| Erythromycin (false) | 120 (83.9) |
| Doxycycline (false) | 135 (94.4) |

The mean knowledge score of the respondents was 15.86 ± 3.34. The lowest correct responses were noted in the questions pertaining to periodontal health while the most correct answers were related to the use of drugs in pregnant patients (Table 1).

The responses to the questions addressing the attitude are shown in Table 2. Most respondents (97.2%) agreed with the importance of maintaining oral health during pregnancy. About 67 (46.9%) dentists expressed concern for treating pregnant patients without acquiring consent from their general practitioners.

Table 3 details various practice considerations of the respondents. About 37.1% of the dentists advised pregnant patients to delay the dental visit until after pregnancy. Scaling and polishing (55.9%) was the most commonly performed procedure by respondents while full mouth radiographs (71.3%) were the most avoided procedure.

**Table 2  Responses for perinatal oral healthcare related attitude items (N = 143).**

| Attitude | Agree N (%) | Not Sure N (%) | Disagree N (%) |
|---|---|---|---|
| Maintaining oral health during pregnancy is important | 139 (97.2) | 3 (2.1) | 1 (0.7) |
| Pregnant women should receive a dental check early in their pregnancy | 138 (96.5) | 4 (2.8) | 1 (0.7) |
| Treatment of periodontal disease during pregnancy positively affects pregnancy outcome | 78 (54.5) | 48 (33.6) | 17 (11.9) |
| Pregnant women are more likely to seek dental care if their antenatal care providers recommend it | 112 (78.3) | 28 (19.6) | 3 (2.1) |
| Antenatal care providers are better able than dentists to counsel pregnant women about oral health | 70 (49) | 28 (19.6) | 45 (31.5) |
| I am concerned about providing treatment to pregnant women without consent from their GPs | 67 (46.9) | 19 (13.3) | 57 (39.9) |
| Currently there is good understanding between ANC providers and dentists regarding dental care for pregnant women | 44 (30.8) | 36 (25.2) | 63 (44.1) |
| My practice is too busy to provide oral health advice for pregnant women | 30 (21) | 12 (8.4) | 101 (70.6) |
| There is insufficient time to advise pregnant women on oral health during a dental visit | 42 (29.4) | 24 (16.8) | 77 (53.8) |
| I have the skills to advise pregnant women on oral health | 117 (81.8) | 21 (14.7) | 5 (3.5) |
| The cost of dental treatment is a barrier to advising pregnant women | 61 (42.7) | 33 (23.1) | 49 (34.3) |
| There is little I can do to affect a pregnant woman's oral hygiene | 42 (29.4) | 18 (12.6) | 83 (58) |
| The link between periodontal disease and preterm birth is too tenuous for me to warn pregnant women about it | 65 (45.5) | 47 (32.9) | 31 (21.7) |
| The link between dental caries in mothers and in babies is too tenuous for me to warn my patients about it | 56 (39.2) | 42 (29.4) | 45 (31.5) |
| I am interested in further information about dental care to pregnant women | 132 (92.3) | 5 (3.5) | 6 (4.2) |
| I am interested in further training to provide dental assessments to pregnant women | 125 (87.4 | 13 (9.1) | 5 (3.5) |
| There is a need for universal guidelines for oral health care during pregnancy for all health professionals | 130 (90.9) | 8 (5.6) | 5 (3.5) |

Only 83 (58%) dentists counseled the patients regarding caries prevention and transmission.

The concern of pregnant women regarding the safety of dental procedures (77.6%) and the lack of concern for oral health during pregnancy (74.1%) were the most common barriers identified in the provision of perinatal oral health care (Fig. 1).

The advice to delay dental visits until after pregnancy was 8 times more likely to be observed among dentists who lacked the knowledge of the importance of oral health during pregnancy ($P = 0.04$, OR = 8.75). Dentists were more likely to consult obstetricians regarding dental procedures when they fear a risk of labour in the dental practice ($P < 0.05$, OR = 3.72) or when they feel they have less time for providing advice in practice ($P < 0.05$, OR = 3.49).

The results of binary logistic regression are shown in Table 4. Dentists who had the knowledge of periodontal disease association with preterm delivery were about four times more likely to treat periodontal disease during pregnancy in the hope of positively affecting pregnancy outcomes ($P = 0.013$, OR = 3.95). Dentists lacking updated knowledge of periodontal disease and its consequences in pregnant women were 2.5 times more likely to not warn the pregnant women about association of periodontal disease with pregnancy ($P = 0.01$, OR = 2.56). Dentists knowing about the association between maternal oral health and childhood decay were more likely to counsel pregnant patients regarding caries prevention ($P > 0.05$, OR = 3.75).

**Table 3 Responses for perinatal oral healthcare related practices of dentists (N = 143).**

| Practices | Always N (%) | Sometimes N (%) | Never N (%) |
|---|---|---|---|
| I discuss the importance of oral health with pregnant women during clinical care | 94 (65.7) | 48 (33.6) | 1 (0.7) |
| I advise pregnant women to delay dental visits until after pregnancy | 53 (37.1) | 81 (56.6) | 9 (6.3) |
| I advise pregnant women to visit dentists during early pregnancy | 83 (58) | 50 (35) | 10 (7) |
| I provide counselling regarding the association of poor periodontal health with negative birth outcomes | 63 (44.1) | 49 (34.3) | 31 (21.7) |
| I provide counselling regarding caries prevention and transmission | 83 (58) | 49 (34.3) | 11 (7.7) |
| I consult obstetricians before/after dental procedures | 39 (27.3) | 73 (51) | 31 (21.7) |
| Types of dental treatment I advise to receive during pregnancy: | | | |
| Routine examination | 119 (83.2) | 22 (15.4) | 2 (1.4) |
| Routine cleaning | 110 (76.9) | 31 (21.7) | 2 (1.4) |
| Periodontal (gum) treatment | 83 (58) | 44 (30.8) | 16 (11.2) |
| Fillings/crowns | 74 (51.7) | 51 (35.7) | 18 (12.6) |
| Routine treatments I undertake: | | | |
| Scaling and root planning | 80 (55.9) | 42 (29.4) | 21(14.7) |
| Radiographs, single periapical | 32 (22.4) | 62 (43.4) | 49 (34.3) |
| Full mouth radiographs | 10 (7) | 31 (21.7) | 102 (71.3) |
| Single tooth extraction | 38 (26.6) | 74 (51.7) | 31 (21.7) |
| Endodontic/root canal treatment therapy | 45 (31.5) | 83 (58) | 15 (10.5) |
| Administrating local anaesthetic | 39 (27.3) | 80 (55.9) | 24 (16.8) |
| Providing nitrous oxide | 12 (8.4) | 28 (19.6) | 103 (72) |
| Extracting tooth impactions | 12 (8.4) | 36 (25.2) | 95 (66.4) |
| Opening and broaching to relieve pain | 74 (51.7) | 59 (41.3) | 10 (7) |
| Incising and draining an abscess | 67 (46.9) | 69 (48.3) | 7 (4.9) |
| Placing a temporary restoration | 68 (47.6) | 69 (48.3) | 6 (4.2) |
| Antibiotics | 31 (21.7) | 84 (58.7) | 28 (19.6) |

# DISCUSSION

As scientific evidence highlighting the importance of perinatal oral health continues to grow, the provision of oral health care for pregnant women becomes more important (*Vergnes & Sixou, 2007*; *Wazir et al., 2019*; *Loesche, 1997*). This study identified a positive attitude of dentists towards the provision of oral health care to pregnant women with 97.2% recognizing its importance. A similar attitude was expressed by dentists in other parts of the world as well (*George et al., 2017*; *Mayberry, Norrix & Farrell, 2017*).

Despite the development of perinatal oral health guidelines and consensus over the safety of various dental procedures in pregnancy, a hesitation to treat pregnant women continues to persist among dentists (*Lee et al., 2010*; *Steinberg et al., 2013*). In this study 37.1% of dentists advised pregnant women to defer dental appointments until after pregnancy which is significantly more than that reported (1.6%) by Australian dentists (*George et al., 2017*). However, this was less than that reported by *Wali et al. (2016)* (47.5%) while investigating a population of Pakistani dentists. These differences can be explained
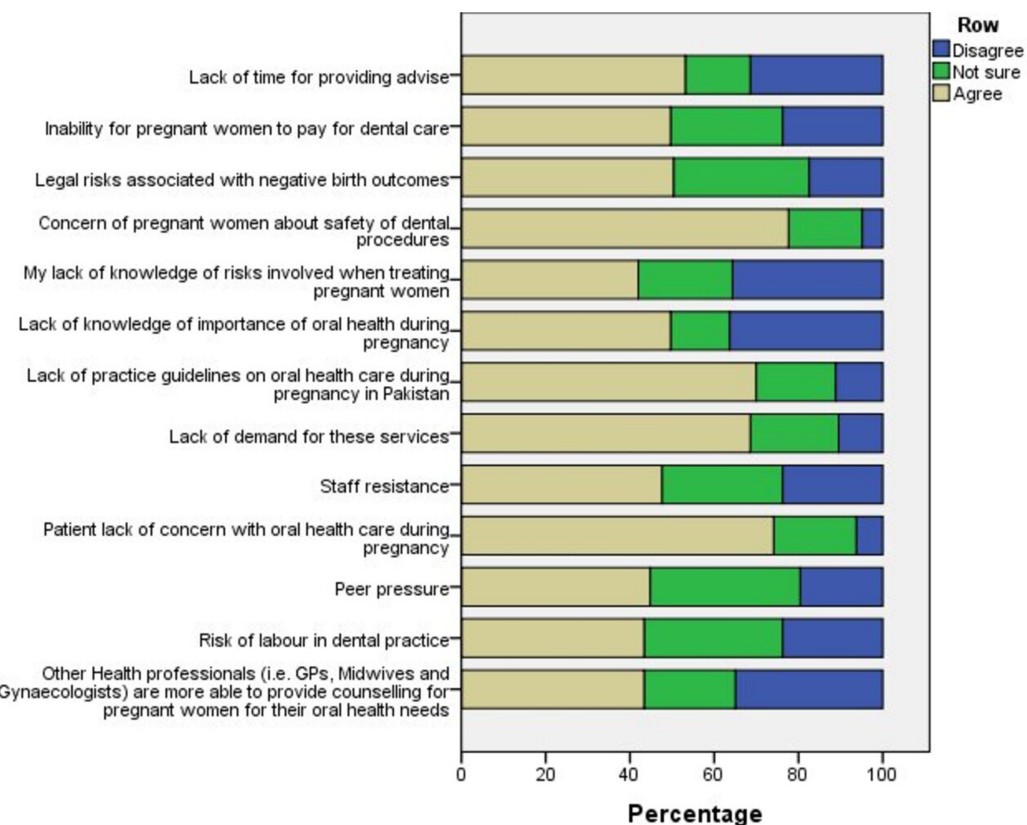

**Figure 1 Description of barriers to perinatal oral health care provision for dentists.**

by the variable adoption of evidence-based practices by dentists practicing in different regions. The literature has explored various reasons behind this hesitation to treat pregnant women. *George et al. (2017)* identified the lack of knowledge of risks associated with the treatment of pregnant women and the concern of treating them without the prior consent of attending general physicians as important predictors for deferring dental appointment. However, our study highlighted that dentists who did not recognize the importance of perinatal oral health were more likely to delay dental visits of pregnant women.

While 42% of dentists in the present study identified the lack of knowledge of risks associated with treating pregnant women as a barrier in offering treatment to pregnant women, 81.8% still believed in having the adequate skill to advise pregnant women on oral health. A similar disparity in the confidence to advise pregnant women and the apprehension of treating them was reported in the studies of *Vieira et al. (2015)*, *George et al. (2017)*, *Huebner et al. (2009)* and *Da Costa et al. (2010)*.

The knowledge of a majority of the dentists participating in the present study conformed to the current evidence-based practices for perinatal oral health. About 88.8% of dentists were aware of the safety of basic dental treatments in pregnancy. These findings are comparable to the reports on Turkish and Pakistani dentists (*Wali et al.,*

Table 4 Association of predictor variables with practice and attitude variables (N = 143).

| Practice or attitude variable | Predictor variables | Odds ratio (95% CI) |
|---|---|---|
| [1]Practice of counselling pregnant patients regarding caries prevention and transmission | Poor maternal oral health can contribute to early childhood decay | 3.749 [0.756–18.580] |
| | The link between dental caries in mothers and in babies is too tenuous for me to warn my patients about it | 0.198 [0.024–1.644] |
| | Lack of practice guidelines on oral health care during pregnancy in Pakistan | 0.816 [0.221–3.022] |
| [2]Practice of counselling pregnant women regarding association of poor periodontal health with negative birth outcomes | Periodontal disease has been associated with the Spontaneous Abortion/Miscarriage | 1.551 [0.481–5.00] |
| | Periodontal disease has been associated with the stillbirth | 0.931 [0.242–3.57] |
| | Periodontal disease has been associated with the preeclampsia | 1.052 [0.319–3.475] |
| | The link between periodontal disease and preterm birth is too tenuous for me to warn pregnant women about it | 2.561 [1.04–6.307]* |
| | Maintaining oral health during pregnancy is important | 1.679 [0.149–18.938] |
| | Lack of practice guidelines on oral health care during pregnancy in Pakistan | 0.470 [0.171–1.294] |
| [3]Attitude variable "treatment of periodontal disease during pregnancy positively affects pregnancy outcome" | Periodontal disease has been associated with the Preterm Delivery | 3.953 [1.61–9.69]* |
| | Periodontal disease has been associated with the Spontaneous Abortion/Miscarriage | 0.791 [0.28–2.22] |
| | Periodontal disease has been associated with the low birth weight | 1.770 [0.70–4.42] |

Notes:
[1] Negelkerke R square = 0.120, (Hosmer and Lemeshow Test, $P = 0.776$)
[2] Negelkerke R square = 0.087, (Hosmer and Lemeshow Test, $P = 0.889$), *$P < 0.05$
[3] Negelkerke R square = 0.234, (Hosmer and Lemeshow Test, $P = 0.24$), *$P < 0.05$

2016; *Ugurlu & Orhan, 2019*). However, dentists from Australia and the United States were overall more knowledgeable regarding the safety of dental procedures in pregnancy (*Bokhari & Khan, 2006*; *George et al., 2011*; *George et al., 2017*). In the current study, knowledge regarding the safety of certain procedures like exposing dental radiographs (31.5%), dental extractions (49%), local anesthesia (51.7%), and root canal treatment (67.85%) was found to be deficient. Similar results were reported in the previous studies (*Vieira et al., 2015*; *Lee et al., 2010*; *George et al., 2017*; *Huebner et al., 2009*; *Wali et al., 2016*; *Leong et al., 2013*). Even though dental radiography is not contraindicated in pregnancy (*Ide & Papapanou, 2013*), several studies have reported similar misinformation of dentists regarding its safety (*Vieira et al., 2015*; *Lee et al., 2010*; *George et al., 2017*; *Huebner et al., 2009*; *Wali et al., 2016*; *Leong et al., 2013*). Most of the dentists (68.5%) in the present study believe that dental radiography is unsafe in pregnancy and which explains why only 22.4% of the dentists agree with the practice of taking periapical radiographs. While the judicious use of dental radiographs during pregnancy is acceptable, it is best to avoid unnecessary radiation exposure and to limit such practices to the second trimester (*Yenen & Ataçağ, 2019*). This can be explained perhaps by the possibility that

most dentists might associate safety of dental radiography with $2^{nd}$ trimester as observed by *Huebner et al. (2009)*. The 43.4% of the dentists who use periapical radiographs "sometimes" in the present study could have perceived a similar trimester associated safety of dental radiography. This can be counterproductive in establishing an accurate diagnosis and delivering appropriate treatment to pregnant women.

Another misinformation is regarding the safety of local anesthetic use in pregnant women as only half of the respondents in the current study correctly regarded them as safe. This is contrary to current evidence wherein lignocaine is safe as a local anesthetic for use in pregnant women (*Steinberg et al., 2013*). A similar trend was reported by *Ugurlu & Orhan (2019)* while *George et al. (2017)* reported a much higher percentage of dentists recognizing the safety of local anesthetics in pregnancy (*George et al., 2017*; *Leong et al., 2013*). The safety of most commonly prescribed drugs was acknowledged by the majority of the dentists in our study which is encouraging as inappropriate use of drugs can have detrimental effects on perinatal health. Surprisingly, the safest drug on the list was amoxicillin and it was identified as safe by relatively fewer respondents (*Naseem et al., 2016*). This can be alarming, as Amoxicillin is the first choice of antibiotics for the management of pregnant patients reporting facial swelling (*Ouanounou & Haas, 2016*).

Perhaps the most severe deficiencies in knowledge were present with respect to the association of the periodontal disease with various pregnancy complications. The same was observed among dentists of other nationalities as well (*George et al., 2017*; *Huebner et al., 2009*; *Leong et al., 2013*). Despite the mixed evidence of an association between perinatal periodontal health and obstetric complications in literature, a positive association has been recognized in economically disadvantaged societies (*Mobeen et al., 2008*; *Han, 2011*; *Offenbacher et al., 2006*). Therefore, in Pakistan it becomes more important for dentists to accept the importance of this association and counsel pregnant women accordingly. It was encouraging to see that dentists who regarded this association as weak were even more likely to advise pregnant women regarding the said association ($P < 0.05$, OR = 2.56). Perhaps the knowledge of preterm birth association with poor periodontal health was encouraging them to have a positive counseling attitude ($P = 0.001$, OR = 3.953). Unlike a periodontal disease, the evidence of perinatal oral health as a risk factor for early childhood caries is more established (*Xiao et al., 2019*). Dentists who knew of the association between poor maternal oral health and early childhood caries were more likely to counsel regarding prevention and transmission ($P > 0.05$, OR = 3.49). However, this disparity between knowledge and practice should be further evaluated due to a lack of statistical significance ($P = 0.20$). Additionally, those who considered this association weak were less likely to counsel pregnant patients ($P > 0.05$, OR = 0.198). The lack of practice guidelines on perinatal oral health care in Pakistan was identified as a barrier in the provision of counseling regarding periodontal health ($P > 0.05$, OR = 0.470) and caries prevention ($P > 0.05$, OR = 0.816).

The results of the present study should be carefully interpreted as there are a few limitations. To begin with, the response rate was low probably due to a large number of items in the surveying instrument. Most experienced dentists have busy routines and might be reluctant to complete long surveys, hence explaining the relatively young age of

the respondents in the study. However, this would result in underreporting of correct knowledge and practice parameters as seen in the present study. Another limitation was the absence of trimester-based distinction in the knowledge and practice questions. This resulted in perhaps many respondents selecting the option "sometimes" while answering practice-related questions. Lastly, the research was conducted in one metropolitan area (Rawalpindi–Islamabad) of Pakistan, therefore a cautious approach is recommended while generalizing the findings of the research. Moreover, a future multicenter/multicity study is recommended.

## CONCLUSIONS

Collectively, the results of our study indicate few gaps in knowledge among some dentists and a need to improve existing attitudes towards perinatal oral health. Moreover, there is a need to address issues like the development of practice guidelines for perinatal oral health care in Pakistan. Dentists failing to recognize the importance of perinatal oral health are more likely to be hesitant in treating pregnant patients. Failing to recognize the link between periodontal disease and obstetric complications increases the possibility of hesitance to counsel pregnant patients regarding the same. The appreciation of the evidence for poor perinatal oral health and risk of early childhood caries increases the likelihood of counseling by dentists on caries prevention.

## ACKNOWLEDGEMENTS

The authors would like to thank Dr. Ajesh George, University of Western Sydney, for permission to use his questionnaire.

### Funding

This work was supported by the Deanship of Scientific Research, Qassim University. The funders had no role in study design, data collection and analysis, decision to publish, or preparation of the manuscript.

### Grant Disclosures

The following grant information was disclosed by the authors:
Deanship of Scientific Research, Qassim University.

### Competing Interests

The authors declare that they have no competing interests.

### Author Contributions

- Muhammad Qasim Javed conceived and designed the experiments, performed the experiments, analyzed the data, prepared figures and/or tables, authored or reviewed drafts of the paper, and approved the final draft.
- Usman Anwer Bhatti analyzed the data, authored or reviewed drafts of the paper, and approved the final draft.

- Arham Riaz performed the experiments, prepared figures and/or tables, and approved the final draft.
- Farooq Ahmad Chaudhary conceived and designed the experiments, authored or reviewed drafts of the paper, and approved the final draft.

## Human Ethics

The following information was supplied relating to ethical approvals (*i.e.*, approving body and any reference numbers):

Institutional Review Committee, College of Dentistry, Riphah International University, Islamabad, Pakistan (iidc/irc/2018/03/07).

## Data Availability

The raw data are available in the Supplemental Files.

## Supplemental Information

Supplemental information for this article can be found online at http://dx.doi.org/10.7717/peerj.12080#supplemental-information.

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
