# Peer review of "Predictors of periodontal and caries related perinatal oral healthcare, investigation of dentists’ practices: a cross-sectional study"

_PeerJ, doi:10.7717/peerj.12080_

## Round 0.1 · original submission · Major Revisions

Dear authors,

Please follow the reviewers' instructions.

You do not have to cite sources specifically described by the reviewers but you could choose other sources describing the oral microbiome in their introduction.

Best regards

Reviewer 1 ·

Basic reporting

A. English is globally acceptable but a deep revision performed by a native English speaker is strongly suggested.

B. Literature references are quite good even if I have to underline that in introduction (when dealing about periodontal disease and its association with Preterm LBW) no literature references to the crucial role of Oral Microbiota are given. I suggest you some references to be added in order to strengthen the global evidence emerging from the manuscript:
1. Characterizing peri-implant and sub-gingival microbiota through culturomics. First isolation of some species in the oral cavity. A pilot study. Pathogens, 2020:9(5): 365;
2. Oral microbiota: Discovering and facing the new associations with systemic diseases. Pathogens, 2020:9(4):313;
3. Relationship between oral microbiota and periodontal disease: A systematic review. Eur Rev Med Pharmacol Sci, 2018:22(18): 5775-5788.

Experimental design

Some relevant issues to be highlighted in this section are:

A. Please report the study design in the title.

B. Association between oral diseases and Preterm LBW and ECC is to be better explained and commented in introduction.

C. The primary aim of the study has to be reported by conforming as much as possible to the PICO format.

D. In materials and methods section the eligibility criteria and methods of selection for study population are to be reported. Why questionnaires have been sent by mail? Why not through e-mail? This could have helped to reach a bigger number of potential respondents.

E. In discussion please report the external validity and generalizability of findings.

Validity of the findings

Some relevant issues to be highlighted in this section are:

A. For what emerges from results about included dentists it is evident that age interval is too short as well as mean experience. This evidence configures a selection bias that has to be disclosed in discussion. The result about Mean and SD of working hours per week is too high (more than 50 hours per week!), please double-check it.

B. Another element to be added when defining the profile of respondents is the geographic area of provenance (e.g. dentists from bigger cities and from rural areas) in order to compare answers from different locations.

C. In results please report also the other significative point emerging from Table 4. This element has also to be discussed in discussion stressing the fact that a potential explanation is that not all dentists are prone to go on with their scientific updating. This hypothesis could also be claimed for explaining the multiple elements about apparent ignorance of a part of respondents that emerge from the discussion.

D. Information about funding is missing, please report.

Additional comments

Dear authors,

All the elements to be changed have been reported in the appropriate boxes.
My general suggestion is to have a first round of major revision in order to solve all highlighted problems.

Regards

Reviewer 2 ·

Basic reporting

English language must be substantially improved for the publication.

Experimental design

Aim of the study
The aim of the study is very superficial and lacks details. Is impossible to understand the first and the secondary outcomes. Please consult STROBE for reporting.

Materials and methods:
Materials and methods are extremely hard to follow, the information is scattered lacks logical sequence or cohesion. Please add subsections (ex. Study design, Study population, Statistical analysis…..) and consult STROBE for reporting.
The statistical analysis section (within the material and method) lacks details. What is the dependent variable? What is the covariable? Did you use different models in the Binomial regression? How did you select the models? Have you made any adjustments? Please add more info.

Validity of the findings

Results
Some analyses in the statistical analysis are not reported in the results (Chi-square and Fisher’s exact test). Please add.

---

## Round 0.2 · accepted · Accept

Dear authors thank you for submitting the revise version of your manuscript.

Best regards

Reviewer 2 ·

Basic reporting

No comment

Experimental design

No comment

Validity of the findings

No comment

Additional comments

Required Reviews Completed